# Intercomparison of aerosol volume size distributions derived from AERONET ground based remote sensing and LARGE in situ aircraft profiles during the 2011-2014 DRAGON and DISCOVER-AQ experiments

Joel S. Schafer[1,3], Tom F. Eck[2,3], Brent N. Holben[3], Kenneth L. Thornhill[4], Luke D. Ziemba[4], Patricia Sawamura[4], Richard H. Moore[4], Ilya Slutsker[1,3], Bruce E. Anderson[4], Alexander Sinyuk[1,3], David M. Giles[1,3], Alexander Smirnov[1,3], Andreas J. Beyersdorf[4,5], and Edward L. Winstead[4]

[1]Science Systems and Applications, Inc. Lanham, MD, USA
[2]Universities Space Research Association, Columbia, MD, USA
[3]NASA Goddard Space Flight Center, Greenbelt, MD, USA
[4]NASA Langley Research Center, Hampton, Virginia, USA
[5]University of California at San Bernardino, CA, USA

**Correspondence:** Joel Schafer (joel.schafer@nasa.gov)

**Abstract.**

Aerosol volume size distributions (VSD) retrievals from the Aerosol Robotic Network (AERONET) aerosol monitoring network were obtained during multiple DRAGON (Distributed Regional Aerosol Gridded Observational Network) campaigns conducted in Maryland, California, Texas and Colorado from 2011 to 2014 . These VSD retrievals from the field campaigns

were used to make comparisons with near simultaneous in situ sampling from aircraft profiles carried out by the NASA Langley Aerosol Group Experiment (LARGE) team as part of four campaigns comprising the DISCOVER-AQ (Deriving Information on Surface conditions from Column and Vertically Resolved Observations Relevant to Air Quality) experiments. For coincident ($\pm$ 1 hour) measurements there were a total of 91 profile-averaged fine mode size distributions acquired with the LARGE Ultra-High Sensitivity Aerosol Spectrometer (UHSAS) instrument matched to 153 AERONET size distributions

retrieved from almucantars at 22 different ground sites. These volume size distributions were characterized by two fine mode parameters, radius of peak concentration ($r_{peak\_conc}$) and VSD fine mode width ($width_{fine\_mode}$). The AERONET retrievals of these VSD fine mode parameters, derived from ground-based almucantar sun photometer data, represent ambient humidity values while the LARGE aircraft spiral profile retrievals provide dried aerosol (RH<20%) values. For the combined multiple campaign data set, the average difference in $r_{peak\_conc}$ was $0.033 \pm 0.035$ $\mu$m (ambient AERONET values were 15.8% larger

than dried LARGE values) and the average difference in $width_{fine\_mode}$ was $0.042 \pm 0.039$ $\mu$m (AERONET values were 25.7% larger). For a subset of aircraft data, the LARGE data were adjusted to account for ambient humidification. For these cases, the AERONET-LARGE averages differences were smaller, with $r_{peak\_conc}$ differing by $0.011 \pm 0.019$ $\mu$m (AERONET values 5.2% larger) and $width_{fine\_mode}$ average differences equal to $0.030 \pm 0.037$ $\mu$m (AERONET values 15.8% larger).

## 1  Introduction

Atmospheric aerosol volume size distribution information is relevant to modeling of radiative transfer, weather processes and human health via air quality concerns [Peng et al. (2018), Li et al. (2015), Sheng et al. (2019), Eilenberg et al. (2018), Gong et al. (2003)]. Interactions of atmospheric aerosols with clouds are highly sensitive to their size distributions [Feingold (2003)].
Current climate models are now able to simulate the full aerosol size distributions and therefore benefit from accurate aerosol size parameterization [Li et al. (2015)]. Geographic and seasonal variability in atmospheric aerosol due to differences in aerosol type and composition were historically difficult to capture globally at high temporal resolutions. The Aerosol Robotic Network (AERONET) global monitoring program provides an opportunity to capture seasonal and diurnal trends in extinction-weighted column integrated aerosol volume size distribution and concentration for ambient atmospheric conditions derived from frequent
sky radiance measurements and spectral aerosol optical depth.

Very few direct comparisons of size distribution between in situ and AERONET retrievals have been published. Eck et al. (2012) summarized a number of region-specific comparison studies focused on both fine and coarse mode. During INDOEX Clarke et al. (2002) computed lognormal fits of volume size distributions for in situ measurements for fine mode pollution acquired by ship and aircraft in the Arabian Sea under high aerosol loading and found average accumulation mode volume peak
radius ranged from 0.17 to 0.18 $\mu$m with computed geometric standard deviations for ship data equal to 1.51 and aircraft data equal to 1.43. These values are similar to the AERONET retrieved averages from 1998 to 2000 in the Maldives (Kaashidhoo) in the same region. For this 2 year observation period the AERONET determined average volume median radius was 0.18 $\mu$m with a width of 1.49 for almucantars taken with AOD at 440nm exceeding 0.4 which agrees well with the Clarke et al. (2002) results. Reid et al. (2005) investigated the agreement of in situ measurements of the volume median radius for smoke from
various distinct regions of biomass burning; Southern Africa, North America (temperate and boreal) and South America) with retrievals from AERONET. For each region, the in situ volume median diameter typically agreed with AERONET retrievals within $\sim$ 0.01 mm. Retrievals of larger radius (sub-micron) aerosol from AERONET almucantars have also compared well with in situ data as detailed in Eck et al. (2010) where Pinatubo stratospheric peak volume radius of $\sim$ 0.56 $\mu$m derived from AERONET retrievals was very similar to the effective radius of 0.53 $\mu$m noted by Pueschel et al. (1994) based on in situ
stratospheric aircraft measurement.

This paper presents a large number of comparisons of multiple fine mode volume size distribution datasets from four US regions for in-situ measurements from repeated aircraft profiles during a series of month-long intensive DISCOVER-AQ campaigns conducted between 2011 and 2014. Given the typical complexity of aircraft campaigns and the fact that validation of AERONET retrieval products is rarely a central campaign goal, this well-coordinated effort resulted in a dataset unique for the
quantity and near simultaneous nature of the comparisons.

## 2  Instrumentation

In the summer of 2011 AERONET deployed more than forty Cimel sun photometers in the Baltimore-Washington, DC region as part of DRAGON (Distributed Regional Aerosol Gridded Observational Network) campaign of which five were located at

DISCOVER-AQ aircraft profile sites [Holben et al. (2018)]. The AERONET DRAGON mesoscale network was comprised of automatic sun/sky radiometers distributed on a roughly 10km grid (covering an area of approximately 60km x 120km; average distance between sites= 9.9km) which operated continuously for more than 2 months. The duration of the DISCOVER-AQ aircraft measurement interval (profiles) was 4-5 weeks in length for each campaign. The subsequent campaigns in CA, TX and

CO were less densely instrumented than MD with each using approximately 15 ground sites (5-6 of which were profile sites) in the San Joaquin Valley (California), Houston Metro (Texas) and Colorado Front Range with average distance between sites ranging from 20-25km. The DRAGON ground networks for each campaign are depicted in Figure 1; vertical spiral profile sites used are shown in red.

The AERONET DRAGON campaign was concurrent with the NASA sponsored DISCOVER-AQ air quality experiment

which performed daily research flights concentrating on repeated multiple daily profile measurements of gaseous and particulate pollution over typically 5-6 primary sun photometer sites. The number of flights days for each campaign ranged from 10 to 16 with atmospheric conditions ranging from very low aerosol optical depth (AOD) with low column water vapor (AOD 500nm < 0.05; CWV < 1cm) to hazy and humid (AOD 500nm $\sim$ 0.81; CWV > 4.5cm).

A complete description of the sun photometers used is provided by Holben et al. (1998). All sun photometers at profile

sites had narrow bandpass filters with central wavelengths of 340, 380, 440, 500, 675, 870, 940, 1020 and 1640nm which cover the visible and near infrared spectrum. Eck et al. (1999) describes the uncertainty in aerosol optical depth which varies with wavelength (larger in the ultraviolet) and ranges from $\sim$ 0.01-0.021 for sun photometers during deployment. Direct solar irradiance is measured at each wavelength (FOV 1.2°) as well as radiance from the sky in both the principal plane ($\sim$ 9 times daily) and the solar almucantar ($\sim$ 8 times daily) which is taken at four wavelengths (440, 675, 870 and 1020nm). The

almucantar procedure records sky radiance every 0.5°-1° close to the position of the sun (azimuth angles from 3.5°-8°) and with decreasing angular frequency further from the sun (angular steps increasing from 2°-20°). Both aerosol optical depth measurements and sky radiance from almucantars are input to inversion code used to routinely produce AERONET retrievals of volume size distribution, phase function, real and imaginary component of refractive index, effective radius and single scattering albedo [Dubovik and King (2000); Dubovik et al. (2002); Dubovik et al. (2006)]. The AERONET retrieval products

have quality controls applied based on Holben et al. (2006). Both aerosol optical depth (AOD) and alumucantar retrievals are from the version 3 dataset (version 3 data released in January 2018) [Giles et al. (2019)].

In-situ aerosol properties were measured on the NASA P-3B aircraft by the NASA Langley Aerosol Group (LARGE) team using a suite of instruments to characterize ambient aerosol optical and microphysical properties [Beyersdorf et al. (2016)]. A DMT Ultra-High Sensitivity Aerosol Spectrometer (UHSAS) calibrated with ammonium sulfate was utilized for particle sizing

measurements. Dry ammonium sulfate aerosol particles were generated and size classified with a differential mobility analyzer before being introduced into the UHSAS to determine the true measurement calibration. Typically, the UHSAS is calibrated with NIST-traceable polystyrene latex spheres that have a real refractive index of 1.59 that is not realistic for naturally-occurring atmospheric aerosols. Shingler et al. (2016) conducted a comprehensive study of aerosol dry refractive index for a variety of air mass types encountered during the NASA SEAC4RS field campaign. They observed that the real part of the refractive index

for dry particles was fairly constant at between 1.52-1.54 for all air mass categories, which is consistent with the real part of

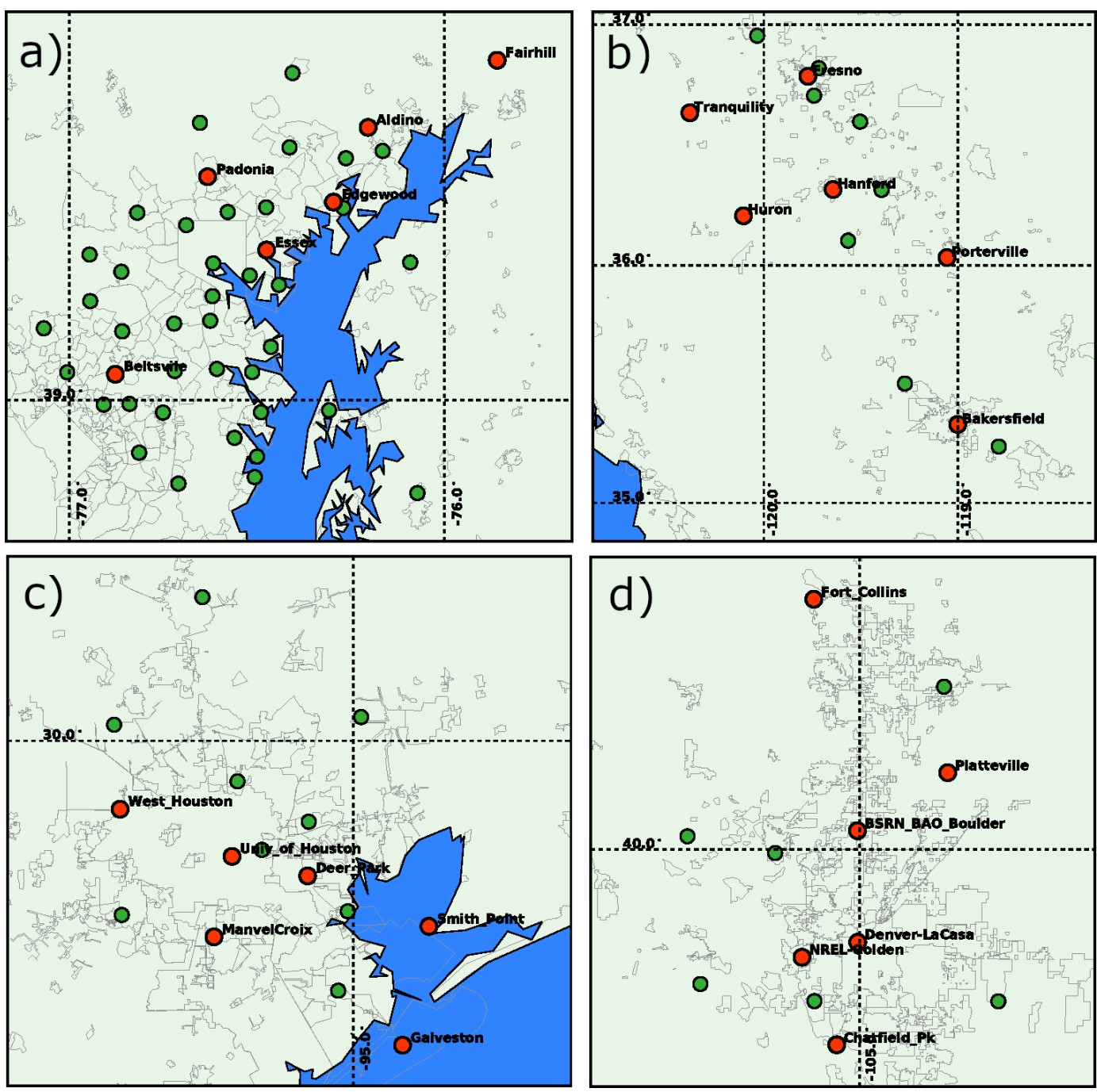

**Figure 1.** The AERONET ground networks are shown for each campaign. Profile sites used are shown in red with urban boundaries overlaid in gray. a) Maryland b) California c) Texas d) Colorado

the refractive index of ammonium sulfate reported as 1.521 [Shingler et al. (2016)]. Hygroscopic growth can also affect the refractive index of the aerosol, but is not a factor in this measurement since the air is heated and dried (via RAM effects) upon entering the cabin via the isokinetic inlet. Aerosol optical measurements were made with a pair of TSI-3563 3-wavelength integrating nephelometer (TSI, Inc. model 3563) and a 3-wavelength Radiance Research Particle Soot Absorption Photometer

(PSAP, Radiance Research). The tandem nephelometers were run with and without humidification to find the dry scattering (approximately 20% relative humidity) and humidified scattering coefficients (approximately 80% relative humidity). Scattering at ambient relative humidity was then calculated based on a single-parameter monotonic growth factor – gamma [Gassó et al. (2000)]. Scattering coefficients at 450, 550 and 700 nm were corrected for truncation errors according to Anderson and Ogren (1998). Absorption coefficients were measured at 470, 532 and 660 nm, and corrected for filter scattering according to

Virkkula (2010).

## 3   Method

The Langley Aerosol Research Group Experiment (LARGE) aircraft team carried out measurements during spiral profiles at altitudes which could range from less than 150m up to greater than 5000m above ground level (ABL) depending on the profile site. On many flight days, these profiles were repeated at each site 3-4 times with individual profiles (ascent or descent)

lasting 5 to 15 minutes. AERONET Cimel sun photometers were operated at each ground profile site as well as numerous secondary locations in the vicinity. The profiles used in this study were limited to those where sampling heights covered the majority of the normal aircraft height range to provide an adequately representative column sample. This typical profile depth varied with campaign, e.g. the San Joaquin valley sampling had lower maximum heights due to the prevalent shallow winter boundary layer. Most sun photometer retrievals products (though not volume size distribution) only reach low uncertainty for

high aerosol loading ($\geq 0.4$ at 440nm). Almucantars also must be taken with large solar zenith angle (SZA > 50) and have low residual error (typically < 5%, increasing to a max of 8% at high AOD) for the retrieval calculation to meet Level 2 quality control criteria. Additionally a minimum number of sky radiance measurements in each of four scattering angle bins must meet symmetry requirements in comparison of the two sides (symmetric about the solar azimuth angle) of the almucantar scan [Holben et al. (2006)]. This last criterion effectively requires that the almucantar be taken during cloud-free or minimally

cloudy conditions. The LARGE aircraft measurements provided continuous number size distribution data at 1 sec sampling rate for the particle radius range from 0.03 to 0.5 $\mu$m (in 79 size bins for MD, 25 bins for subsequent campaigns). The UHSAS data are acquired as particle number counts per dlogDp ($\#/cm^3$) so these bins were geometrically converted to total aerosol volume ($\mu m^3/cm^3$) in a unit cm box representing an equivalent radius bin size and then multiplied by the specific flight depth for each profile interval for comparison with column integrated volume size distributions ($\mu m^3/\mu m^2$) from AERONET surface

retrievals, which require no assumption of column aerosol height. Each LARGE in situ sample measurement was individually weighted by the coincident scattering $\sigma_{SP}$ at 550nm and averaged for the profile according the following equation.

$$VSD_{(weighted\_mean)} = \frac{\sum_{i=0}^{N} \left[ \frac{\sigma_{SP(sample)}}{\sigma_{SP(profile\_mean)}} * VSD_{(sample)} \right]}{N} \qquad \text{N= number of 1 sec samples in profile} \qquad (1)$$

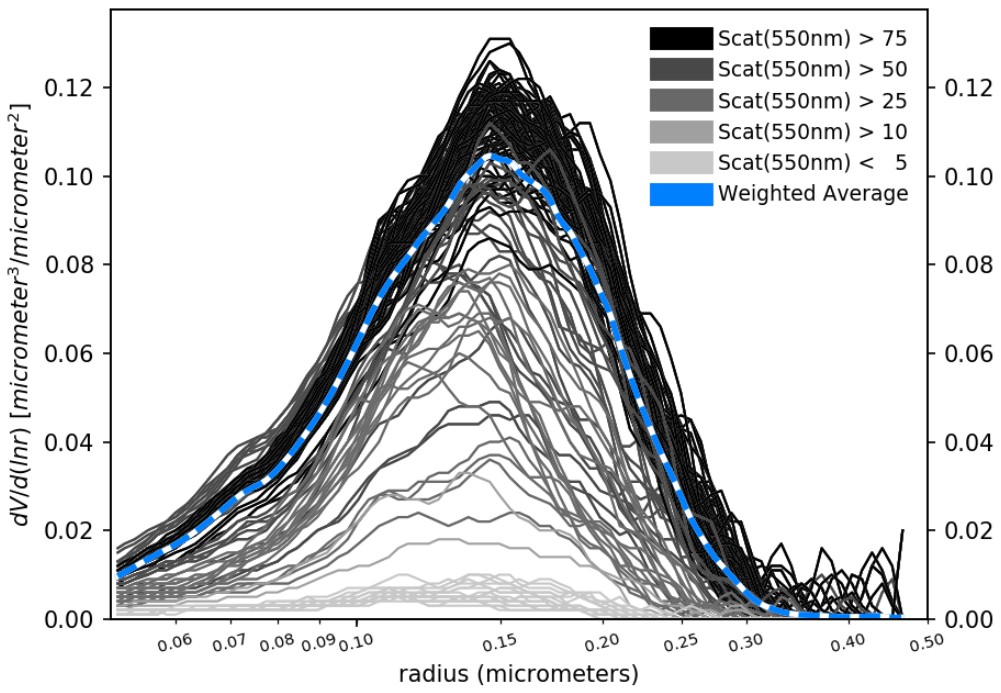

**Figure 2.** An example of the individual LARGE aerosol 1 sec sample size distributions from UHSAS (and scattering weighted average) from a full 22 minute aircraft profile at Essex on July 5th, 2011 (11:30 GMT)

Without such weighting by scattering, the aerosol VSD samples taken within the main aerosol layer would be weighted equally with samples from areas with negligible aerosol where measurement accuracy is diminished. An example of the complete set of individual 1 sec samples from the UHSAS instrument during a full profile can be seen in Figure 2. The flat curves on the bottom of the plot are from the higher altitude samples when aerosol concentrations were very low.

5      The metrics typically employed to characterize AERONET size distributions are the volume median radius of the fine mode (VMR$_f$) and the standard deviation or width of the fine mode distribution, sigma, which are standard AERONET inversion products. These parameters were not computed for the LARGE data in this comparison since the upper limit of the UHSAS sampler (0.5 micrometer) is often much less than the calculated fine mode upper boundary of particle radius for the Cimel retrieval algorithm which can vary between 0.439 and 0.992 $\mu$m, dependent upon the inflection point between fine and

10    coarse modes. Therefore, alternative metrics were used: radius of peak concentration ($r_{peak\_conc}$) and full width half-maximum ($width_{fine\_mode}$) and the size distribution data from the Cimel sun photometer was restricted to the same upper radius boundary as the LARGE data for optimal comparability. These alternative metrics were well correlated with the standard AERONET retrieval products of VMR$_f$ ($r_{peak\_conc}$: $r^2$= 0.88) and sigma (distribution width: $r^2$= 0.63) indicating that they are fair representations of these parameters. Correlation of these metrics with the standard AERONET retrieval products was weaker for

**Table 1.** Average differences and standard deviations (in $\mu$m) in $r_{peak\_conc}$ and $width_{fine\_mode}$ between AERONET and LARGE derived values for all campaign comparisons with no humidification adjustments applied. Also, AERONET average $r_{peak\_conc}$ and $width_{fine\_mode}$ for each campaign, AERONET-LARGE differences as % of average values, average profile maximum RH and the number of comparisons are shown.

| Campaign | $\Delta\, r_{peak\_conc}$ | $\Delta\, width_{fine\_mode}$ | $\overline{r_{peak\_conc}}$ | $\Delta\, peak(\%)$ | $\overline{width_{fine\_mode}}$ | $\Delta\, width(\%)$ | $\overline{RHmax}$ | N |
|---|---|---|---|---|---|---|---|---|
| MD | 0.054±0.027 | 0.059±0.032 | 0.233 | 23.2% | 0.211 | 28.2% | 74.6% | 18 |
| CA | 0.044±0.039 | 0.053±0.044 | 0.189 | 23.0% | 0.170 | 31.0% | 68.7% | 71 |
| TX | 0.016±0.020 | 0.026±0.028 | 0.148 | 10.6% | 0.127 | 20.8% | 72.0% | 37 |
| CO | 0.014±0.020 | 0.026±0.020 | 0.143 | 9.6% | 0.123 | 20.9% | 54.3% | 27 |

larger AOD which would be expected since these conditions would normally be associated with the cases where a larger particle radius upper boundary of the fine mode was determined by the AERONET retrieval. Although the UHSAS instrument size range does not necessarily always encompass the entire fine mode, parameterization as these alternative metrics does allow for an effective comparison of the peak volume radius and size distribution width using similarly calculated AERONET column

averaged metrics. AERONET retrievals acquired within $\pm$ 1 hour of a complete aircraft profile were identified for all four DISCOVER-AQ campaigns in order to compare VSD fine mode metrics derived from AERONET retrievals with those from UHSAS sampling data taken by the LARGE aircraft team.

## 4 Aerosol Volume Size Distribution Comparisons

AERONET Level 2 (quality-assured) Version 3 inversions (N=153) derived from AERONET almucantar protocols were

matched with concurrently ($\pm$ 1 hour) measured LARGE aircraft profile sampling sequences (N=91). These were compiled to generate statistics for observed AERONET-LARGE average differences and standard deviations (in micrometers, $\mu$m) of the computed peak radius of concentration and VSD fine mode width for the four DISCOVER-AQ campaigns which are presented in Table 1. Here the LARGE measured size distributions are only for dried aerosol data (RH<20%) as compared to retrieved ambient aerosol VSD from AERONET.

Campaign-averaged differences in $r_{peak\_conc}$ (AERONET-LARGE) for the four regional campaigns ranged from 0.014 $\mu$m to 0.054 $\mu$m and average volume size distribution width differences ranged from 0.026 $\mu$m to 0.059 $\mu$m. Figure 3 depicts color-coded AERONET-derived differences between campaigns as a function of average profile relative humidity where the marker size is proportional to the aerosol optical depth (440nm) acquired by the Cimel sun photometer within $\pm$ 30 minutes of the retrieval. Because AERONET retrievals inherently represent ambient humidity atmospheric conditions, it might be

expected that the size distributions would be shifted to larger sizes for these retrievals relative to LARGE, particularly for more humidified conditions. For the radius of peak concentration there is indeed a significant increasing trend in AERONET-LARGE differences in the Maryland ($r^2$=0.7) and California ($r^2$= 0.5) data with higher relative humidity likely due to hygroscopic growth of particle size. From hygroscopicity observations during the Maryland campaign, Ziemba et al. (2013) found that

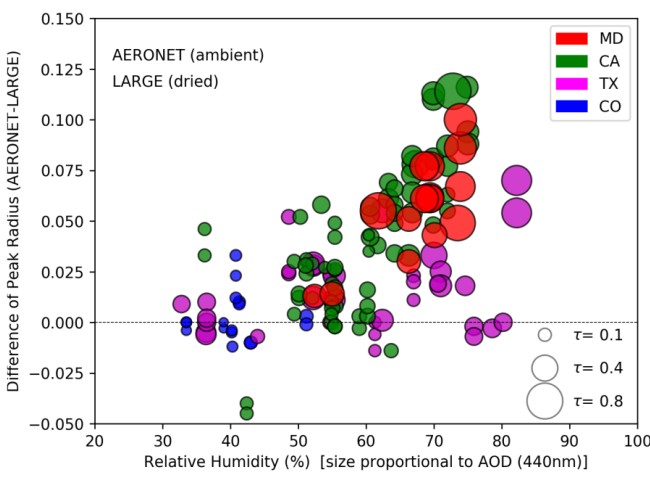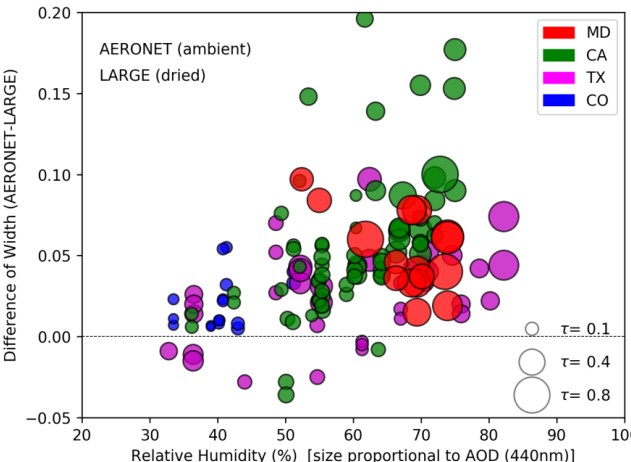

**Figure 3.** Differences in $r_{peak\_conc}$ [left] and $width_{fine\_mode}$ [right] between AERONET (ambient aerosol) and LARGE (dried aerosol) versus profile relative humidity. Marker is size is proportional to coincident aerosol optical depth (440nm) from AERONET

on average, liquid water contributed up to 43% to the ambient extinction coefficient during the study. Note that in the case of California data, the occurrence of shallow layer fog events would not be evident in the column average RH values for the profile since the aircraft did not sample within the fog layer and never recorded the highest relative humidity conditions. Despite hot and humid conditions during the Texas campaign, there was no significant increasing trend of AERONET-LARGE

differences of $r_{peak\_conc}$ with relative humidity and generally the Texas data showed smaller discrepancies than CA and MD at comparable AOD amounts. The Texas profile sites are proximal to many petroleum refining and chemical production facilities that could produce aerosols that are distinct from MD and CA in terms of composition, hygroscopicity and/or amount of aging. The degree of agreement of VSD fine mode width was not strongly associated with relative humidity for any campaign.

    Comparisons for the Colorado campaign show the smallest differences which is consistent with typically small AOD and

low relative humidity along the Front Range of the Rocky Mountains. Despite the low to moderate AOD and low column water vapor during the California winter campaign, there are typically larger differences between AERONET and LARGE retrievals than for the Maryland and Texas comparisons, particularly for fine mode width. A possible expanation is that some of these cases are associated with incomplete sampling of the aerosol layer by the aircraft profile for days with particularly shallow boundary layers. The vertical distribution of atmospheric aerosol was quite distinct for the California dataset in that

75% of the aircraft profiles used in these comparisons had the majority of the aerosol below 500m, with 5 profiles where > 90% of aerosol was in this narrow altitude range. The average minimum sampling height in California was 170 meters while the average altitude of peak scattering was only 110 meters higher at 280 meters, so these profiles may be more at risk of missing a moderate portion of the aerosol layer situated below the minimum profile sample altitude. For comparison, the average altitude of maximum aerosol scattering observed by aircraft was $\sim$ 1km in Maryland and $\sim$ 1.2km in Texas and

neither of these regions acquired any profiles with such shallow aerosol layers as observed in California. The average fraction

of aerosol scattering contributed by the lowest 500m of the atmosphere in Maryland was only 6% (15% for the Texas profiles), whereas in California the lower layer aerosol comprised on average 64% of the total aerosol scattering in the profile. We note however that while most California profile locations had higher minimum altitudes, there were a few sites that relied on missed approach aircraft maneuvers to acquire more complete sampling of the atmosphere. We therefore compared the agreement of AERONET-LARGE VSD metrics for these missed approach locations with the agreement typically observed at the more common sites with shallower profiles. While we found some cases where there was a significant increase in fine mode peak size at a low level (well below the minimum altitude of most CA sites which would therefore would be missed by aircraft sampling), we did not observe a general tendency for better agreement for the deep profile locations. This doesn't preclude the possibility that aircraft profiles from sites with higher minimum altitudes are occasionally missing distinctly different aerosol to which the columnar observations from AERONET are sensitive, but this factor may not be a primary contributor to AERONET-LARGE VSD metric differences. Another known factor is that the California campaign comparisons were also complicated by high frequency of thick morning fog in the San Joaquin Valley during this winter campaign which often generated fog-processed aerosol that changed rapidly in time. It has previously been observed that the influence of persistent fog conditions on aerosol properties can produce significant changes for the hygroscopic fraction and that these changes can also persist beyond the dissipation of the fog or cloud [Eck et al. (2012)]. This type of modification event was documented by Eck et al. (2012) for AERONET inversions of volume size distributions at Fresno, California, a location also included in this DISCOVER-AQ study, where fog-processed aerosols exhibited very large fine radius, even larger than humidified aerosols at high relative humidity and high column water vapor amounts in Maryland and Texas. Both of these factors, typically shallow aerosol layers and frequent and persistent morning fog, could lead to greater potential for disagreement between LARGE and AERONET measurements for the California campaign. Histograms of the differences in these parameters (for dried aerosol in the LARGE data) are presented for the combined data from all campaign in Figure 4.

## 4.1 Comparisons With Humidification Adjustment of LARGE Volume Size Distributions

The effect of aerosol humidification on observed differences in the aircraft and AERONET comparisons was estimated using a simple particle growth factor for each UHSAS sample from LARGE. The growth factor depends on the differences between dry and ambient scattering using auxiliary data from on-board nephelometer and Particle Soot Absorption Photometer (PSAP) data. It is well known that a significant fraction of the aerosol volume consists of condensed water under elevated relative humidity conditions, which needs to be accounted for when comparing the AERONET-retrieved volume at ambient humidification to the LARGE size distributions measured at dry (<20%RH) conditions. A correction for this was made by scaling the LARGE dry size distributions by an effective growth factor, g, that is derived from coincident scattering and absorption measurements and Mie Theory following the methodology of Sawamura et al. (2016). Implicit in the use of an effective growth factor is that the aerosol is internally mixed and its composition does not vary with size – i.e., the entire size distribution can be shifted by a single scale factor, g. The first step in the growth factor computation is to use the measured dry aerosol size distribution and measured dry scattering and absorption coefficients to compute the dry aerosol refractive index using Mie Theory. This dry refractive index is then used with the measured humidified scattering coefficient and Mie Theory to iteratively solve for

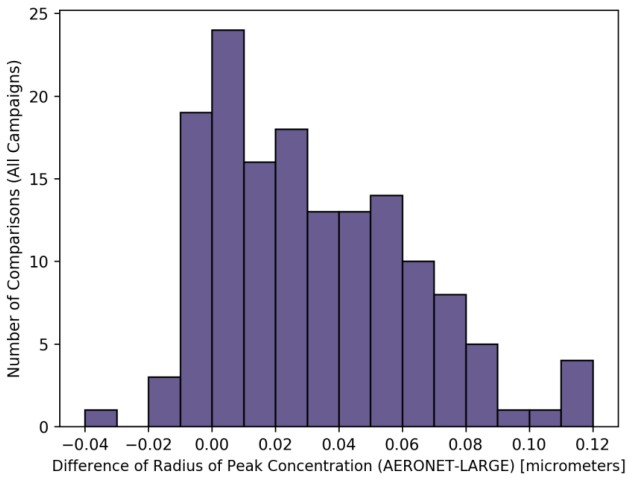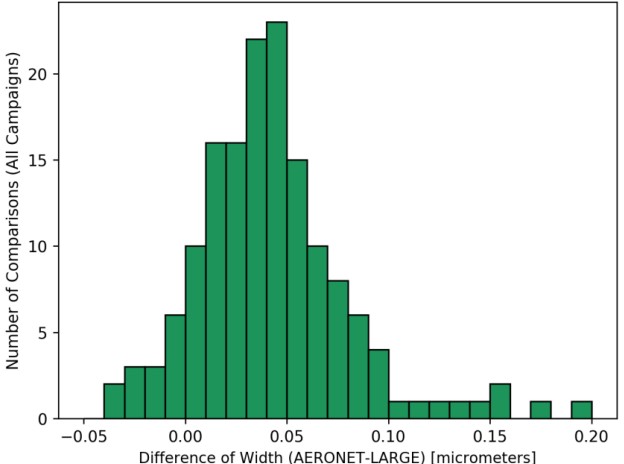

**Figure 4.** Histograms of all differences of $r_{peak\_conc}$ [left] and $width_{fine\_mode}$ [right] between AERONET (ambient aerosol) and LARGE (dried aerosol) for combined campaign comparisons

the effective growth factor, assuming that the humidified aerosol refractive index is the volume-weighted average of the dry particle refractive index and that of pure water (1.33 - 0i). Sawamura et al. (2016) found that the measured in situ aerosol volume and the measured ambient HSRL vertically-resolved retrieval aerosol volume were in excellent agreement once the effective growth factors were used to convert the dry in situ measurements to ambient humidification values. This size invariant
growth factor, scaled by scattering coefficient and averaged for each profile, was used to scale the particle sizes for the dried aircraft aerosol volume size distribution to better approximate the same columnar ambient VSD as provided by the AERONET retrievals.

Not all comparison cases had complete data for all three component sensors (UHSAS, PSAP and nephelometer) so a full profile of growth factor adjusted VSD was not always possible to generate. The effect of this correction can be seen in Table
2 where mean differences VSD statistics are shown for the subset of comparisons where a humidity adjustment could be computed. Comparisons between AERONET and LARGE data for both uncorrected and humidity adjusted LARGE data are shown as well as the range of computed growth factor values applied to each campaign. The application of a growth factor adjustment was observed to reduce the average discrepancy in $r_{peak\_conc}$ by ∼0.02-0.03 $\mu$m and in $width_{fine\_mode}$ by ∼0.01-0.02 $\mu$m.

The fraction of profiles that had sufficiently complete growth factor computations varied from only 15% of comparisons for Maryland to 57% for Texas. Colorado has no growth factor adjusted cases, though this region would have the least impact from humidification of aerosol due to consistently low RH, as suggested by the relatively good agreement between AERONET retrievals at ambient RH versus LARGE measurements for dried aerosol seen in Table 1. Additional details are provided in Table 3 (as in Table 1) but restricted to only the subset of comparisons where the growth factor could be computed.

**Table 2.** For the subset of comparison where humidification adjustment of LARGE data was possible, average differences (in $\mu$m) in $r_{peak\_conc}$ and $width_{fine\_mode}$ between AERONET and LARGE derived values are shown for uncorrected and growth factor corrected (GF) cases. Also, the range of growth factor applied for each campaign subset and the number of comparisons are shown.

| Campaign | $\Delta\, r_{peak\_conc}$ | $\Delta\, r_{peak\_conc(GF)}$ | $\Delta\, width_{fine\_mode}$ | $\Delta\, width_{fine\_mode(GF)}$ | GF Range | N |
|---|---|---|---|---|---|---|
| MD | 0.037 | 0.009 | 0.065 | 0.043 | 1.17-1.20 | 3 |
| CA | 0.045 | 0.024 | 0.065 | 0.048 | 1.09-1.27 | 24 |
| TX | 0.014 | 0.000 | 0.023 | 0.012 | 1.03-1.24 | 27 |

**Table 3.** Average differences and standard deviations (in $\mu$m) in $r_{peak\_conc}$ and $width_{fine\_mode}$ between AERONET and LARGE derived values for only cases that include computed humidification adjustments applied to LARGE data. Also, AERONET average $r_{peak\_conc}$ and $width_{fine\_mode}$ for each campaign, and AERONET-LARGE differences as % of average values are shown.

| Campaign | $\Delta\, r_{peak\_conc}$ | $\Delta\, width_{fine\_mode}$ | $\overline{r_{peak\_conc}}$ | $\Delta\, peak(\%)$ | $\overline{width_{fine\_mode}}$ | $\Delta\, width(\%)$ |
|---|---|---|---|---|---|---|
| MD | 0.009$\pm$0.015 | 0.043$\pm$0.027 | 0.191 | 4.5% | 0.184 | 23.3% |
| CA | 0.024$\pm$0.016 | 0.048$\pm$0.041 | 0.183 | 13.1% | 0.173 | 27.9% |
| TX | 0.000$\pm$0.015 | 0.012$\pm$0.023 | 0.148 | 0.2% | 0.124 | 10.1% |

Numerous examples of the VSD from LARGE and AERONET from the three campaigns for cases with computed growth factors are shown in Figure 5. The corresponding vertical profiles of RH are presented in Figure 6. These depict at least one comparison from 13 sites (on 7 different days) with comparisons where humidification factor could be computed for the LARGE data. The AODs (440nm) associated with this set of comparisons ranged from 0.11 to 0.80 (mean: 0.26). The agreement between concurrent VSDs from AERONET and LARGE is normally improved using the growth factor adjusted data for each of the 3 campaigns for which it could be generated.

Note that the agreement of the magnitude of the AERONET and LARGE fine mode volume concentration is often notably poorer than that for the $r_{peak\_conc}$ and $width_{fine\_mode}$ comparisons. The uncertainty of volume concentration for fine mode aerosols is closely related to the uncertainty in the real part of refractive index, AERONET retrievals of which are sensitive to both measurement noise and instrument offset. This is due to low retrieval sensitivity to the real part of the refractive index, and due to measurements being affected not only by possible instrumental offsets (such as instrument calibration) but also affected by other somewhat random factors such as atmosphere inhomogeneity. The variability in the retrieved real part of refractive index is counterbalanced by the variability in the retrieved volume concentration of the fine mode in the AERONET inversion algorithm (Sinyuk et al. 2019, in preparation). Due to somewhat random variability in retrieved real part of refractive index the variability in the retrievals of volume concentration is also random. This may partly explain the good agreement between AERONET retrieved volume concentration and in situ volume concentration in some cases and not in others. The inversion algorithm will reliably provide a highly accurate fit of extinction AOD (within 0.01 at four wavelengths) plus directional sky radiance distributions for each almucantar. It should be mentioned that in the case of inversion of spectral AOD only, uncertainty in the real part of refractive index affects both volume concentration and volume median radius. However, adding

spectral measurements of sky radiances as in AERONET retrievals provides additional constraints for the retrievals of aerosol size thus making them more stable than in the case of inversion of spectral AOD only.

The effect of adding the humidification correction can be directly observed by comparing the difference in VSD metrics for only the subset of cases with corresponding humidified growth factor (GF) adjusted data. For these cases (54 comparisons from 3 campaigns), the combined multi-campaign average of peak radius differences between AERONET and LARGE decreased from 0.029$\mu$m to 0.011$\mu$m and the $width_{fine\_mode}$ difference averages decreased from 0.044 $\mu$m to 0.030 $\mu$m, due to application of humidification growth factors to the LARGE data. The Maryland and Texas campaigns showed the greatest improvement (largest reductions in differences) with very small average differences in peak concentration radius (both < 0.01 $\mu$m) when incorporating this simplified humidification assumption. The average difference for the California campaign (N=27) was reduced from 0.045 $\mu$m to 0.024$\mu$m. As a percentage of the average observed AERONET peak concentration radius this humidification adjusted subset has AERONET-LARGE differences that range from negligible on average for Texas (0.2%) to 13.1% for the California campaign. The best agreement in $width_{fine\_mode}$ parameter was observed for the Texas campaign where the AERONET retrieved width parameter was found to be on average 0.012 $\mu$m larger than the humidity adjusted aircraft data which amounts to 10% of the mean value of AERONET VSD width from the AERONET retrievals. The other two campaigns considered here had average $width_{fine\_mode}$ differences that were greater than that noted for Texas (MD: 0.043 $\mu$m; CA: 0.048 $\mu$m). This may be due in part to the much larger average $width_{fine\_mode}$ of these two campaigns ($\sim$ 0.18 $\mu$m) compared with that in Texas (0.12 $\mu$m) though the difference as a percent of average campaign $width_{fine\_mode}$ were also larger (23-25%).

The $width_{fine\_mode}$ differences (AERONET-LARGE) decrease (for the growth factor adjusted subset) with campaign averaged differences decreasing for each campaign (Texas: 0.023 $\mu$m to 0.012 $\mu$m; MD: 0.065 $\mu$m to 0.043 $\mu$m; California: 0.064 $\mu$m to 0.048 $\mu$m). For the humidity adjusted dataset, 95% comparisons of the radius of peak concentration agreed within $\pm$ 0.05 $\mu$m while 83% of comparisons of the $width_{fine\_mode}$ of the VSD agreed within $\pm$ 0.05 $\mu$m. The small number of cases of larger disagreement in $width_{fine\_mode}$ were all from the California campaign which again may reflect incomplete sampling of the full aerosol layer for days with the shallow wintertime boundary layer typical of the region or potentially extreme growth of fine mode particles in the layer affected by fog in some cases (Eck et al., 2012). Figure 7 depicts the VSD statistic differences as in Figure 3 but for only the subset of comparisons with humidification adjustment and additionally, the corresponding histograms are seen in Figure 8.

Whereas many AERONET retrieval products such as as imaginary refractive index and single scattering albedo (SSA) require larger AOD (AOD 440 > 0.4) for adequate aerosol absorption signal, it was believed that the volume size distribution did not have similar minimum AOD thresholds for valid determination. However, this had not been empirically verified until this study. With regard to this criterion, the agreement of aircraft and sun photometer was found to have no penalty for conditions of relatively low aerosol loading, at least to the levels measured during these field campaigns. Indeed the mean differences in both peak radius and size distribution width were at a minimum for the lowest AOD cases with smaller standard deviations. For the lowest AOD quartile of the comparison set (AOD 440: 0.09-0.15) the average difference in $r_{peak\_conc}$ (AERONET-LARGE) was only 0.011 $\pm$ 0.003 $\mu$m compared to the largest quartile (AOD440: 0.27-0.8) average difference of 0.025 $\pm$ 0.008 $\mu$m.

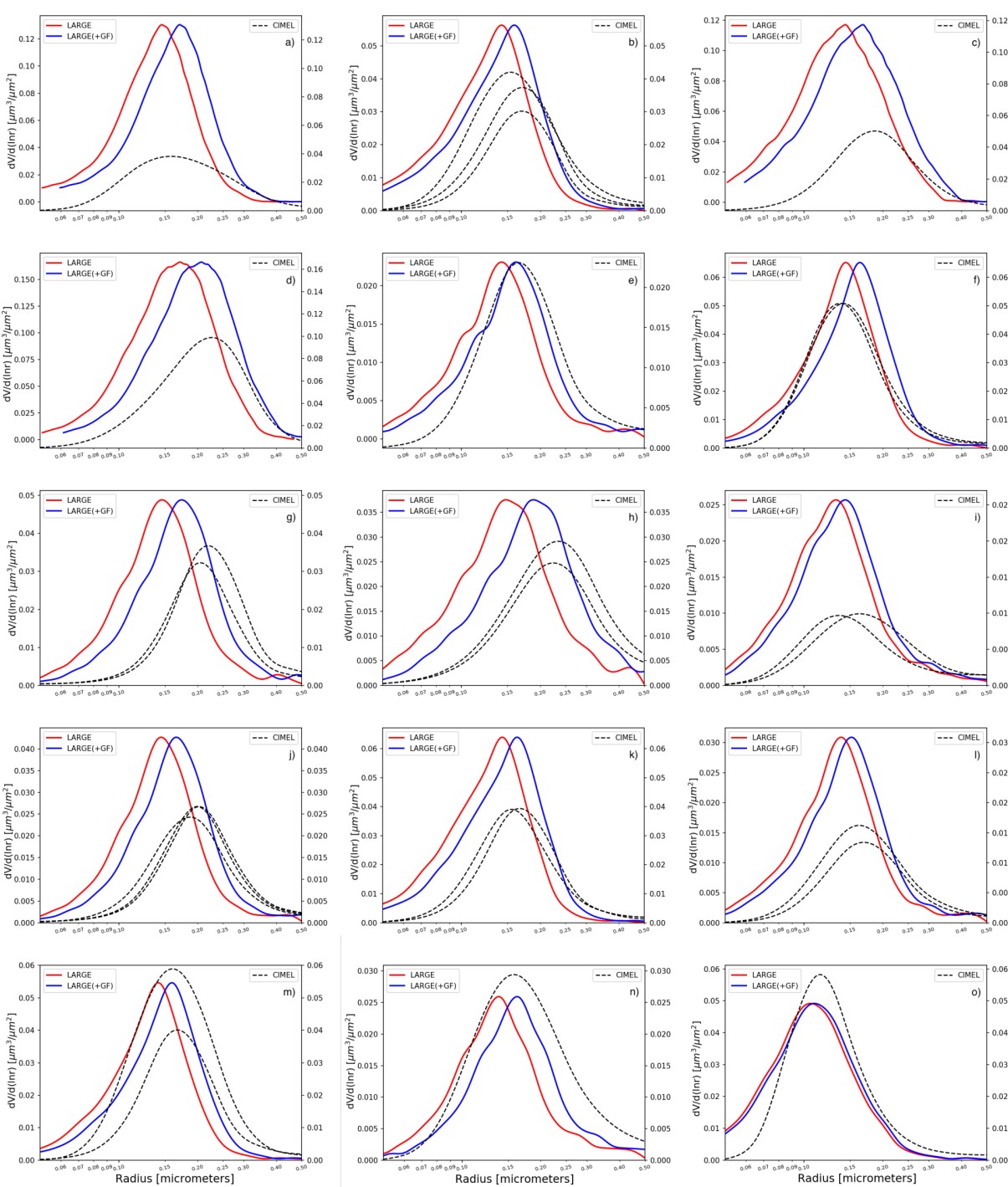

**Figure 5.** AERONET VSD (dash) comparisons with LARGE (red) and humidified VSD (blue) a) Aldino(MD) [2011-07-05] b) Deer_Park(TX) [2013-09-13] c) Edgewood(MD) [2011-07-05] d) FairHill(MD) [2011-07-29] e) Fresno(CA) [2013-02-01] f) Galveston(TX) [2013-09-13] g) Hanford(CA) [2013-02-01] h) Hanford(CA) [2013-02-04] i) Huron [2013-01-31] j) Huron(CA) [2013-02-01] k) ManvelCroix(TX) [2013-09-13] l) Porterville(CA) [2013-01-31] m) Smith_Point(TX) [2013-09-13] n) Tranquility(CA) [2013-02-01] o) West_Houston(TX) [2013-09-25]

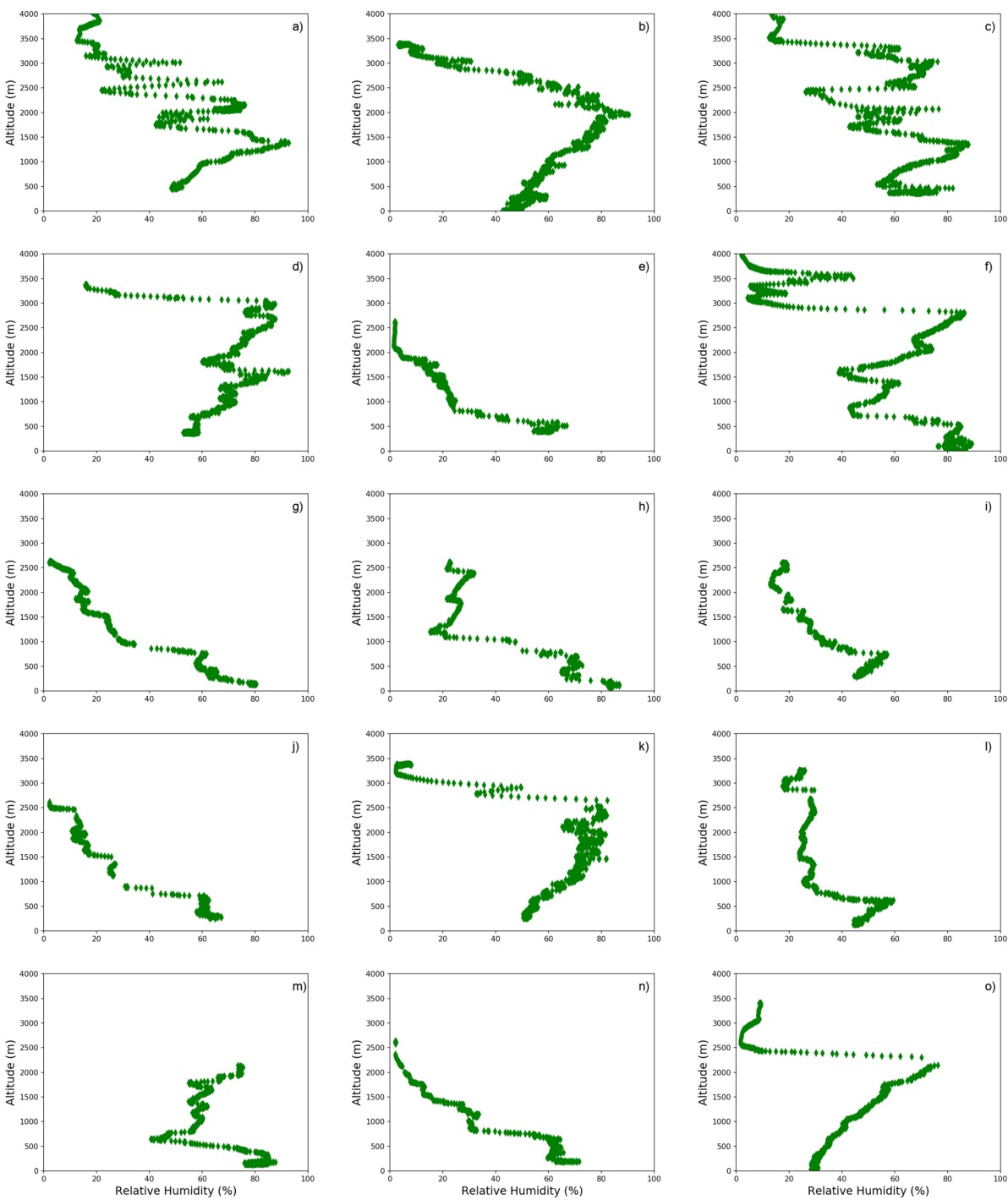

**Figure 6.** Vertical profiles of relative humidity a) Aldino(MD) [2011-07-05] b) Deer_Park(TX) [2013-09-13] c) Edgewood(MD) [2011-07-05] d) FairHill(MD) [2011-07-29] e) Fresno(CA) [2013-02-01] f) Galveston(TX) [2013-09-13] g) Hanford(CA) [2013-02-01] h) Hanford(CA) [2013-02-04] i) Huron [2013-01-31] j) Huron(CA) [2013-02-01] k) ManvelCroix(TX) [2013-09-13] l) Porterville(CA) [2013-01-31] m) Smith_Point(TX) [2013-09-13] n) Tranquility(CA) [2013-02-01] o) West_Houston(TX) [2013-09-25]

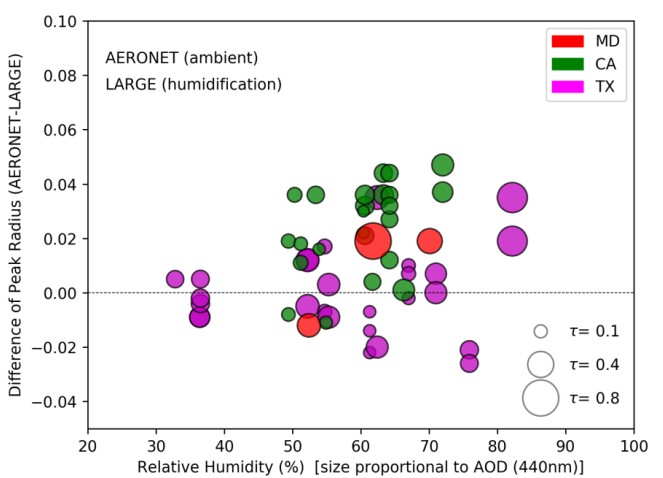
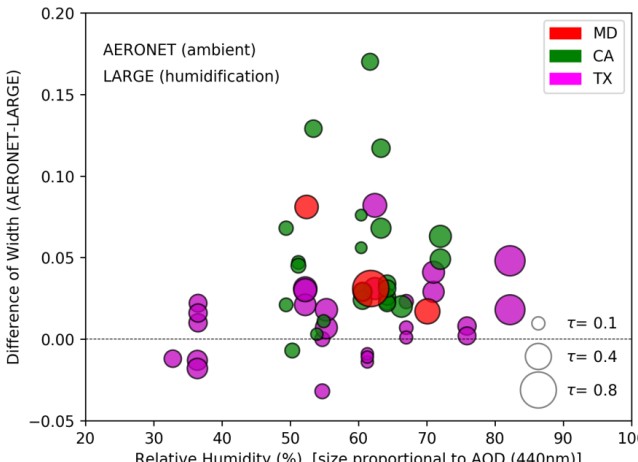

**Figure 7.** Differences in $r_{peak\_conc}$ [left] and $width_{fine\_mode}$ [right] between AERONET (ambient aerosol) and LARGE (with humidification adjustment) versus profile relative humidity. Marker is size is proportional to coincident aerosol optical depth (440nm) from AERONET

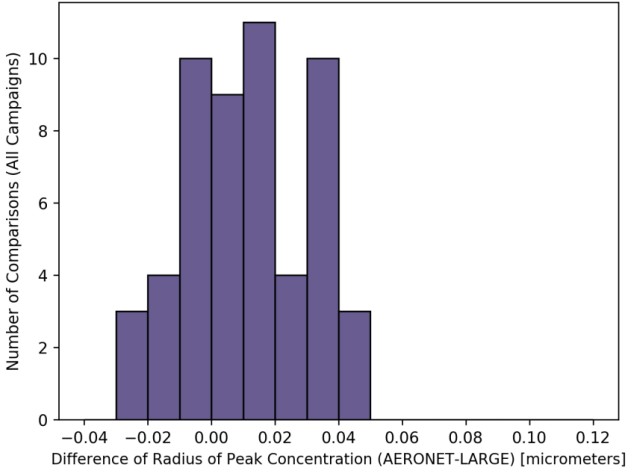
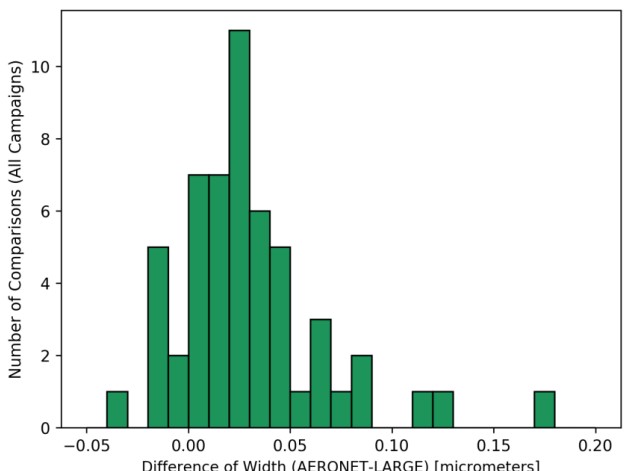

**Figure 8.** Histograms of all differences of $r_{peak\_conc}$ [left] and $width_{fine\_mode}$ [right] between AERONET (ambient aerosol) and LARGE (with humidification adjustment) for combined campaign comparisons

The low AOD comparisons may benefit in part from the fact that these conditions are more commonly associated with lower relative humidities. As such they might be expected to manifest less disparity between measured (dry) VSD from LARGE and retrieved (ambient) VSD from AERONET especially for hydrophilic aerosol species despite our efforts to approximate and correct for this humidification effect. The corresponding AOD quartile average differences for $width_{fine\_mode}$ were also

better for the low AOD comparison set ($0.009 \pm 0.003$ $\mu$m) than the highest quartile ($0.019 \pm 0.008$ $\mu$m). The good agreement in the lowest quartile between aircraft and sun photometer retrievals of both $r_{peak\_conc}$ (mean relative differences of 7.8%) and $width_{fine\_mode}$ (7.2%) with small standard deviations strongly suggests that these retrievals are generally stable even at relatively low aerosol optical depths.

## 5  Conclusions

The DRAGON/DISCOVER-AQ campaigns represent the most extensive comparison of AERONET fine mode column integrated volume size distribution retrievals with in-situ aircraft vertical profile size distribution measurements. These experiments provided a rare opportunity to coordinate multiple instrumented aircraft profiles with AERONET almucantar retrievals at 22 ground sites in Maryland, California, Texas and Colorado during four distinct month-long campaigns (acquired during North American winter, summer and fall seasons) from 2011-2014. Two aerosol fine mode particle size parameters derived from

AERONET and LARGE in situ measurement profiles for the four campaigns (radius of peak concentration, $r_{peak\_conc}$ and volume size distribution width, $width_{fine\_mode}$) were found to generally agree well for both parameters with the overall average difference (AERONET-LARGE; no humidification adjustment to LARGE) for $r_{peak\_conc}$ equal to $0.033 \pm 0.035$ $\mu$m and $0.042 \pm 0.039$ $\mu$m for $width_{fine\_mode}$. When a subset of aircraft data were adjusted to account for the effect of ambient humidity on the dried aerosol measurements, these comparisons had smaller combined campaign averaged differences of

$r_{peak\_conc}$ $0.011 \pm 0.019$ $\mu$m while $width_{fine\_mode}$ average difference were also less ($0.030 \pm 0.037$ $\mu$m) for cases where humidification adjustments were possible. These comparisons were made over a wide range of aerosol optical depths (AOD (440nm) ranging from 0.09 to 0.8) with the smallest AERONET-LARGE differences of both $r_{peak\_conc}$ and $width_{fine\_mode}$ found at lower AOD levels. For the comparisons made using humidification adjusted LARGE data, larger average differences of $r_{peak\_conc}$ and $width_{fine\_mode}$ were observed for the California campaign which was possibly a result of aircraft profiles

which did not sample the full aerosol column and/or the occasional effect of cloud-processed aerosol during numerous several regional fog events.

*Acknowledgements.* The AERONET project is supported by the Radiation Sciences Program (NASA) and the EOS project office (NASA). All AERONET data used in this paper are available at "https://aeronet.gsfc.nasa.gov/cgi-bin/webtool_aod_v3" (AOD) and "https://aeronet.gsfc.nasa.gov/cg bin/webtool_inv_v3" (Retrieval products).

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

**List of Figures**

**List of Tables**