# Peer review of "Intercomparison of aerosol volume size distributions derived from AERONET ground based remote sensing and LARGE in situ aircraft profiles during the 2011-2014 DRAGON and DISCOVER-AQ experiments"

_Atmospheric Measurement Techniques, 2019_

## Referee Comment (RC1) · Anonymous Referee #1 · 21 May 2019

The work "Intercomparison of aerosol volume size distributions derived from AERONET ground based remote sensing and LARGE in situ aircraft profiles during the 2011-2014 DRAGON and DISCOVER-AQ experiments" by Schafer et al. shows a comparison between the Aerosol volume size distributions (VSD) obtained by AERONET with near simultaneous in situ sampling from aircraft profiles acquired with the LARGE UltraHigh Sensitivity Aerosol Spectrometer (UHSAS). Due to the characteristics of UH-SAS instrument, the comparison is focus on the fine mode, and particularly, on the

radius of the peak concentration and fine mode width, which are almost equivalents to the AERONET standard products for the fine mode: volume median radius and standard deviation from volume median radius. The average difference in the radius peak is 0.033  $\mu m$  and this difference decreases to 0.011  $\mu m$  when the effects of ambient humidification (hygroscopic growth) are accounted.

The value of the paper resides in the difficulty to conduct this type of validations with highly equipped aircrafts and coordinated with ground-based measurements. Overall, the manuscript is quite clear and good presented though there are some parts that are not completely well described, and some results are, in my opinion, not sufficiently justified. Therefore, my recommendation is to accept the paper after some major revisions.

Major remarks

- Page 3 line 34 and Page 8 line 15: the way growth-factor is estimated and how is applied to correct the VSD is not clearly presented on the paper. I reckon that some ideas can be gained in the study by Gasso et al. (2000), but due to the importance of this particular issue in the present work, I would strongly suggest including the corresponding explanations here. This may clarify the role of scattering and absorption coefficients measured by the nephelometers and PSAP. On the other hand, the authors could also consider adding a table with the wavelengths used and the aerosol parameters measured or/and retrieved by the different instruments in the study. For instance, the reader needs to guess that the absorption coefficients are measured by PSAP instrument, however, I think that it could be clearly stated in the manuscript as well as the utility of those measurements.
- In the page 7, the authors say that the relatively large differences found for California winter campaign cannot be attributed to the hygroscopic growth since the humidity was too low. They suggest that the explanation could be associated to
the high presence of aerosols in low atmospheric layers. My question here: is there any trend indicating that the VSD for heights smaller than 170 m present a peak in smaller radii? In other words, if we assume some continuity in the aerosol vertical distribution, and we observe the VSD at 280 m (or higher) and those at 170 m, is any tendency observed towards smaller radii?

- Page 11 line 2. The authors suggest that sensitivity to the fine mode concentration in AERONET standard inversion is worse than for the radius and standard deviation of the fine mode. I think that the sentence should be either reformulated or well referenced. For instance, that result cannot be inferred from the study "Accuracy assessments of aerosol optical properties retrieved from Aerosol Robotic Network (AERONET) Sun and sky radiance measurements" by Dubovik et al. 2000.
- Page 12 line 14. Just analyzing the figure 5, it is difficult to believe that there is any variation for the width  $_{fine-mode}$  when applying the growth factor. Moreover, the authors claim that there is an improvement for this parameter in all the campaigns: Texas, from 0.023  $\mu m$  to 0.012  $\mu m$ , MD, from 0.065  $\mu m$  to 0.043  $\mu m$  and California from 0.064  $\mu m$  to 0.044  $\mu m$ . However, if we look at the tables 1 and 2, we find that the improvement only occurs in Texas: the values shown in table 1 for MD and California are 0.048  $\mu m$  and 0.043  $\mu m$ , respectively, hence quite similar to those obtained in table 2 (0.043  $\mu m$  and 0.044  $\mu m$ ). I would think that the variations in the width  $_{fine-mode}$  are related to the elimination of some data in the second study, due to the impossibility of estimating the growth factors in some of the profiles.

Minor remarks

• Page 3, line 3: It's not clear why only the duration of the first campaign is mentioned (2 months) and it is omitted for the ones in California, Texas and Colorado. Interactive comment

This fact could explain the differences in the number of measurements in Table 1.

- Page 3, line 19: I reckon that the expression: ...*further from the sun (2°-10°)* should be changed by ...*further from the sun (1°-20°)*. Note that the standard angles for almucantar measurements includes 7° and 8° (both larger than 6°) and from 100°, the measurements are done only each 20°.
- Page 3, line 28: The following sentence is not clear to me, please revise: Although the UHSAS instrument only covers most of the fine mode size range (radius: 0.05-0.5 μm), it does allow for an assessment of the agreement of peak volume radius, size distribution width over a moderately large range of AODs.
- Page 4, caption figure 1: I think that Colorado and Texas are reversed.
- Page 4, line 18: In the previous page we find a minimum of 0.05  $\mu m$  and here somehow is shifted to 0.03  $\mu m.$
- Page 4, line 20: Although I think that it may be related to the Extinction/Scattering efficiency factors for the typical radius measured in the work, it is not clear why the authors considered the scattering at 500 nm to average the size distribution.
- Page 5, figure 2: Somehow the idea that figure 2 wants to show is not clear using the same color for all the measured size distribution. I suggest representing each size distribution using a color-map considering the scattering coefficient at 500nm (used as a weighing factor for the averaged size distribution). Also, it would be helpful to show the averaged size distribution in this figure.
- Page 5, line 1: I think that the use of "our" here is a bit confusing. I think it would be better to use AERONET standard retrieval products.

---

## Referee Comment (RC2) · Anonymous Referee #2 · 3 Jun 2019

This manuscript presents an interesting intercomparison of aerosol volume size distributions between AERONET and in-situ aircraft profiles. There is an inherent difficulty when comparing different measurement techniques, but this kind of exercise is necessary for an appropriate assessment of different aerosol measurements and products. The manuscript is clear and well written, but figures quality could be improved (font size, proper labelling of axes and units, etc). It benefits from a considerable amount of aircraft profiles that enhance the significance of the comparison. The manuscript is

suitable for AMT and could be accepted after major revisions.

Major comments:

Section 2: The authors should include a more detail description of the in-situ instruments, particularly the UHSAS. This is as important as the AERONET data in this paper and should be describe in detail (including measurement principle, calibration, possible issues with this type of measurement, references of previous intercomparison of UHSAS with other size distribution instruments, etc). Concerning the UHSAS, some major points that should be discussed include:

- How the calibration with ammonium sulphate (AS) might affect the measurements. The authors state that the instrument is calibrated with AS, but the ambient aerosol may have a different refractive index which will affect the retrieve size of the particles. This is a common issue in aerosol optical counters, and the retrieved diameters can be corrected accordingly to the "real" refractive index. This can be a major source of discrepancy depending on the predominant aerosol type and should be taken into account.

See for example Pío, C. A., J. G. Cardoso, M. A. Cerqueira, A. Calvo, T. V. Nunes, C. A. Alves, D. Custódio, S. M. Almeida, and M. Almeida-Silva (2014), Seasonal variability of aerosol concentration and size distribution in Cape Verde using a continuous aerosol optical spectrometer, Front. Environ. Sci., 2, 15, doi:10.3389/fenvs.2014.00015.

- The authors directly talk about volume size distribution but the UHSAS measures number size distribution, so a comment on the conversion from number to volume should be included.

Section 3: The methodology section should include a specific subsection dedicated to the retrieval of the GF from the nephelometer tandem+PSAP data explaining how the GF is retrieved and how it is applied to the measured size distribution from the UHSAS. What is the range of GF retrieved?

Section 4: Ambient relative humidity profiles should be included in the manuscript, stating at least maximum RH encountered in the profiles. This is important to understand the effect of hygroscopic correction, and could be more useful than CWV in Figure 3. How the difference of peak radius and width change as a function of maximum or median RH in the profile?

Minor comments

Page 7, line 3: remove "best quality", I don't think this is necessary...

Table 1: +/- standard deviation? State it in the table caption. Also, adjust the number of significant figures according to the +/- value.

Page 7, line 10: "... is often correlated with higher relative humidity in these regions due to hygroscopic growth..." -> This sentence, as it is written, it is not a result from this study since this is not clearly seen in Figure 3. In my opinion, it is really speculative, for the MD campaign, there are only 3 data points in Figure 3 (so it is not possible to infer any kind of trend) and for the Texas campaign it is also difficult to see a clear trend of increasing difference with AOD and CWD...

Page 7, line 13: Totally agree with this statement, but this should be the same for the radius. The association is not clear for neither of them.

Table 2: The average difference in peak radius and width for these same cases but without applying the hygroscopic growth correction should be included in the table for comparison. Table 2 is not directly comparable with Table 1 because of the cases included are different.

Figure 5: The discussion is focused in the peak radius and width of the size distribution, but looking at Figure 5 there are cases in which the volume concentrations agree well between in-situ and AERONET and others than do not agree. The authors could comment on that.

СЗ

---

## Author Comment (AC1) · 9 Jul 2019

The comment was uploaded in the form of a supplement:
https://www.atmos-meas-tech-discuss.net/amt-2019-90/amt-2019-90-AC1-
supplement.pdf

---

## Author Response (AR1)

The authors thank Referee #1 for the helpful review which has helped to clarify and improve this paper. Please see our replies below.

Page 3 line 34 and Page 8 line 15: the way growth-factor is estimated and how is applied to correct the VSD is not clearly presented on the paper. I reckon that some ideas can be gained in the study by Gasso et al. (2000), but due to the importance of this particular issue in the present work, I would strongly suggest including the corresponding explanations here. This may clarify the role of scattering and absorption coefficients measured by the nephelometers and PSAP. On the other hand, the authors could also consider adding a table with the wavelengths used and the aerosol parameters measured or/and retrieved by the different instruments in the study. For instance, the reader needs to guess that the absorption coefficients are measured by PSAP instrument, however, I think that it could be clearly stated in the manuscript as well as the utility of those measurements.

This description has been significantly expanded with the following:

The effect of aerosol humidification on observed differences in the aircraft and AERONET comparisons was estimated using a simple particle growth factor for each UHSAS sample from LARGE. The growth factor depends on the differences between dry and ambient scattering using auxiliary data from on-board nephelometer and Particle Soot Absorption Photometer (PSAP) data. It is well known that a significant fraction of the aerosol volume consists of condensed water under elevated relative humidity conditions, which needs to be accounted for when comparing the AERONET-retrieved volume at ambient humidification to the LARGE size distributions measured at dry (<20% RH) conditions. A correction for this was made by scaling the LARGE dry size distributions by an effective growth factor, g, that is derived from coincident scattering measurements and Mie Theory following the methodology of Sawamura et al. (2016) Implicit in the use of an effective growth factor is that the aerosol is internally mixed and its composition does not vary with size - i.e., the entire size distribution can be shifted by a single scale factor, g. The first step in the growth factor computation is to use the measured dry aerosol size distribution and measured dry scattering and absorption coefficients to compute the dry aerosol refractive index using Mie Theory. This dry refractive index is then used with the measured humidified scattering coefficient and Mie Theory to iteratively solve for the effective growth factor, assuming that the humidified aerosol real refractive index is the volume-weighted average of the dry particle refractive index and that of pure water (1.33 - 0i). Sawamura et al. found that the measured in situ aerosol volume and the measured ambient HSRL vertically-resolved retrieval aerosol volume were in excellent agreement once the effective growth factors were used to convert the dry in situ measurements to ambient humidification values. This size invariant growth factor, scaled by scattering coefficient and averaged for each profile, was used to scale the particle sizes for the dried aircraft aerosol volume size distribution to better approximate the same columnar ambient VSD as provided by the AERONET retrievals.

In the page 7 the authors say that the relatively large differences found for California winter campaign cannot be attributed to the hygroscopic growth since the humidity was too low. They suggest that the explanation could be associated to the high presence of aerosols in low atmospheric layers. My question here: is there any trend indicating that

**the VSD for heights smaller than 170 m present a peak in smaller radii? In other words, if we assume some continuity in the aerosol vertical distribution, and we observe the VSD at 280 m (or higher) and those at 170 m, is any tendency observed towards smaller radii?**

While the column water vapor was relatively low for the winter California campaign, we didn't indicate that the relative humidity was low. We also noted (page 8, page 12) that hygroscopic particle growth due to persistent fog such as was intermittently observed in the San Joaquin valley was a plausible source of larger particles near the surface for several sites. There were numerous fog events over large areas of the study region where relative humidity was at a maximum and large fine mode particle radius consistent with fog processing of aerosol has been documented during this campaign with both ground-based observations and HSRL measurements. For instance, the time series of retrievals from the Porterville site on Feb 4, 2013 which showed significant temporal trends in size distribution (decreasing radius) following the dissipation of fog. We have also added the average maximum relative humidity observed for the profiles in each regional campaign to Table 1.

In this paper, we had speculated that some of the cases of larger disparity in VSD metrics for the California campaign might be due to the aircraft sampling missing a significant portion of the lower atmosphere when the aerosol layer was guite shallow. We looked at this further in response to your question. While most profile locations had higher minimum altitudes, there were a few sites that relied on missed approaches (false landings at an airport) to acquire more complete sampling of the atmosphere. We examined the LARGE/AERONET comparisons for these missed approach locations with the comparisons from the more common sites with less deep profiles. While we found some cases where there was a significant increase in fine mode peak size at a low level (well below the minimum flight altitude of most CA sites which would therefore would be missed by aircraft sampling), we did not observe a general tendency for better LARGE/AERONET VSD metrics for the complete, deep profiles. This doesn't preclude the possibility that aircraft profiles from sites with higher minimum altitudes are occasionally missing humidity enhanced aerosol to which the columnar observations from AERONET are sensitive. However, we now feel that this scenario is not a primary cause of the greater disagreement and we have indicated this in the paper.

Page 11 line 2. The authors suggest that sensitivity to the fine mode concentration in AERONET standard inversion is worse than for the radius and standard deviation of the fine mode. I think that the sentence should be either reformulated or well referenced. For instance, that result cannot be inferred from the study "Accuracy assessments of aerosol optical properties retrieved from Aerosol Robotic Network (AERONET) Sun and sky radiance measurements" by Dubovik et al. 2000.

The uncertainty of volume concentration for fine mode aerosols is closely related to the uncertainty in the real part of refractive index. AERONET retrievals of the real part of refractive index are sensitive to both measurement noise and instrument offset. Dubovik et al (2000) discusses only the effect of instrumental offsets on aerosol inversions. However observed variability in the retrievals of the real part is stronger than suggested in Dubovik et al (2000). This is due to both low retrieval sensitivity to the real part of the refractive index, and due to measurements being affected not only by possible instrumental offsets (such as

instrument calibration) but also affected by other somewhat random factors such as atmosphere inhomogeneity.

The variability in the retrieved real part of refractive index is counterbalanced by the variability in the retrieved volume concentration of the fine mode in the AERONET inversion algorithm (Sinyuk et al. 2019, in preparation). The inversion algorithm will reliably provide a highly accurate fit of extinction AOD (within 0.01 at four wavelengths) plus directional sky radiance distributions for each almucantar while introducing a potential additional source of uncertainty to the volume concentration and real refractive index retrievals. Due to somewhat random variability in retrieved real part of refractive index the variability in the retrievals of volume concentration is also random.

This description has been elaborated in the article text.

Page 12 line 14. Just analyzing the figure 5, it is difficult to believe that there is any variation for the width fine-mode when applying the growth factor. Moreover, the authors claim that there is an improvement for this parameter in all the campaigns: Texas, from 0.023  $\mu$ m to 0.012  $\mu$ m, MD, from 0.065  $\mu$ m to 0.043  $\mu$ m and California from 0.064  $\mu$ m to 0.044  $\mu$ m. However, if we look at the tables 1 and 2, we find that the improvement only occurs in Texas: the values shown in table 1 for MD and California are 0.048  $\mu$ m and 0.043  $\mu$ m, respectively, hence quite similar to those obtained in table 2 (0.043  $\mu$ m and 0.044  $\mu$ m). I would think that the variations in the width fine-mode are related to the elimination of some data in the second study, due to the impossibility of estimating the growth factors in some of the profiles.

With a size-invariant percentage shift (growth factor) applied the shape of the VSD will not change but the width will, which may not be readily apparent on the log scale plots of figure 5. Since a percentage change is larger in absolute terms at the larger end of the VSD range, and since the growth factor is always shifting the LARGE VSD distribution towards larger sizes, the effect of the humidification adjustment is to increase the computed VSD width.

The improvement noted (reduction in AERONET/LARGE differences of VSD width) is based on only the subset of data where growth factor adjustment was applied in order to isolate the effect of the correction, i.e. the same data with and without any adjustment. This fact was noted at the start of this section but it has now been reiterated just prior to this discussion of the effect of humidification on width comparison. Also, we now include a new table that clarifies this. Table 2 provides the AERONET/LARGE comparisons (for only the subset where humidification could be applied) both with and without humidification adjustment applied so the effect of the growth factor correction can be directly observed.

**Minor remarks**

**• Page 3, line 3: It's not clear why only the duration of the first campaign is mentioned (2 months) and it is omitted for the ones in California, Texas and Colorado.**

The 2 month duration for MD indicates the period during which the AERONET ground network was fully deployed and operational. The aircraft measurements took place over 4-5 weeks for each campaign and this is now clarified in the text.

**• Page 3, line 19: I reckon that the expression: ...further from the sun (2°-10°) should be changed by ...further from the sun (1°-20°). Note that the standard angles for almucantar measurements includes 7° and 8° (both larger than 6°) and from 10°, the measurements are done only each 20°.**

This detail has been clarified:

The almucantar procedure records sky radiance every 0.5°-1° close to the position of the sun (azimuth angles from 3.5°-8°) and with decreasing angular frequency further from the sun (angular steps increasing from 2°-20°).

**Page 3, line 28: The following sentence is not clear to me, please revise: Al- though the UHSAS instrument only covers most of the fine mode size range (radius: 0.05-0.5 $\mu$ m), it does allow for an assessment of the agreement of peak volume radius, size distribution width over a moderately large range of AODs.**

This line has been re-written and also moved to the Method section following the discussion of the alternative size metrics where its implication is more obvious from context.

'Although the UHSAS instrument size range does not necessarily always encompass the entire fine mode, parameterization as these alternative metrics does allow for an effective comparison of the peak volume radius and size distribution width using similarly calculated AERONET column averaged metrics.'

**• Page 4, caption figure 1: I think that Colorado and Texas are reversed.**

Fixed

• Page 4, line 18: In the previous page we find a minimum of 0.05  $\mu$ m and here somehow is shifted to 0.03  $\mu$ m.

Fixed

**• Page 4, line 20: Although I think that it may be related to the Extinction/Scattering efficiency factors for the typical radius measured in the work, it is not clear why the authors considered the scattering at 500 nm to average the size distribution.**

Of the wavelengths at which scattering was measured (450, 550 and 700nm), this wavelength (actually 550nm rather than 500 as originally stated) is the most central to the four wavelengths used in AERONET almucantar retrievals.

**• Page 5, figure 2: Somehow the idea that figure 2 wants to show is not clear using the same color for all the measured size distribution. I suggest representing each size distribution using a color-map considering the scattering coefficient at 500nm (used as a weighing factor for the averaged size distribution). Also, it would be helpful to show the averaged size distribution in this figure.**

Figure 2 has been substantially reworked to show color coding for each sample mapped to scattering coefficient and the weighted average VSD has been added as well.

**• Page 5, line 1: I think that the use of "our" here is a bit confusing. I think it would be better to use AERONET standard retrieval products.**

This sentence has been amended.

\*\* Please note that Richard Moore from the NASA LARGE research group has been added as an author to this revised paper

The authors thank Referee #2 for the helpful review which has helped to clarify and improve this paper. Please see our replies below.

Section 2: The authors should include a more detail description of the in-situ instruments, particularly the UHSAS. This is as important as the AERONET data in this paper and should be describe in detail (including measurement principle, calibration, possible issues with this type of measurement, references of previous intercomparison of UHSAS with other size distribution instruments, etc). Concerning the UHSAS, some major points that should be discussed include:

- How the calibration with ammonium sulphate (AS) might affect the measurements. The authors state that the instrument is calibrated with AS, but the ambient aerosol may have a different refractive index which will affect the retrieve size of the particles. This is a common issue in aerosol optical counters, and the retrieved diameters can be corrected accordingly to the "real" refractive index. This can be a major source of discrepancy depending on the predominant aerosol type and should be taken into account. See for example Pío, C. A., J. G. Cardoso, M. A. Cerqueira, A. Calvo, T. V. Nunes, C. A. Alves, D. Custódio, S. M. Almeida, and M. Almeida-Silva (2014), Seasonal variability of aerosol concentration and size distribution in Cape Verde using a continuous aerosol optical spectrometer, Front. Environ. Sci., 2, 15, doi:10.3389/fenvs.2014.00015.

The following details relevant to UHSAS calibration have been added:

Dry ammonium sulfate aerosol particles were generated and size classified with a differential mobility analyzer before being introduced into the UHSAS to determine the true measurement calibration. Typically, the UHSAS is calibrated with NIST-traceable polystyrene latex spheres that have a real refractive index of 1.59 that is not realistic for naturally-occurring atmospheric aerosols. Shingler et al. (2016) conducted a comprehensive study of aerosol dry refractive index for a variety of air mass types encountered during the NASA SEAC4RS field campaign. They observed that the real part of the refractive index for dry particles was fairly constant at between 1.52-1.54 for all air mass categories, which is consistent with the real part of the refractive index of ammonium sulfate reported as 1.521 (Shingler et al., 2016). Hygroscopic growth can also affect the refractive index of the aerosol, but is not a factor in this measurement since the air is heated and dried (via RAM effects) upon entering the cabin via the isokinetic inlet.

Shingler, T., et al. (2016), Airborne characterization of subsaturated aerosol hygroscopicity and dry refractive index from the surface to 6.5 km during theSEAC4RS campaign, J. Geophys. Res.Atmos., 121, 4188–4210, doi:10.1002/2015JD024498.

**The authors directly talk about volume size distribution but the UHSAS measures number size distribution, so a comment on the conversion from number to volume should be included.**

The following details have been added to the description:

The UHSAS data are acquired as particle number counts per dlogDp (#/cm^3) so these bins were geometrically converted to total aerosol volume ( $\mu$ m^3/cm^3) in a unit cm box representing an equivalent radius bin size and then scaled to the specific flight depth for each profile interval for comparison with column integrated volume size distributions ( $\mu$ m^3/ $\mu$ m^2) from AERONET surface retrievals, which require no assumption of column aerosol height.

**Section 3: The methodology section should include a specific subsection dedicated to the retrieval of the GF from the nephelometer tandem+PSAP data explaining how the GF is retrieved and how it is applied to the measured size distribution from the UHSAS. What is the range of GF retrieved?**

This description has been significantly expanded with the following:

The effect of aerosol humidification on observed differences in the aircraft and AERONET comparisons was estimated using a simple particle growth factor for each UHSAS sample from LARGE. The growth factor depends on the differences between dry and ambient scattering using auxiliary data from on-board nephelometer and Particle Soot Absorption Photometer (PSAP) data. It is well known that a significant fraction of the aerosol volume consists of condensed water under elevated relative humidity conditions, which needs to be accounted for when comparing the AERONET-retrieved volume at ambient humidification to the LARGE size distributions measured at dry (<20% RH) conditions. A correction for this was made by scaling the LARGE dry size distributions by an effective growth factor, g, that is derived from coincident scattering measurements and Mie Theory following the methodology of Sawamura et al.. (2016) Implicit in the use of an effective growth factor is that the aerosol is internally mixed and its composition does not vary with size - i.e., the entire size distribution can be shifted by a single scale factor, g. The first step in the growth factor computation is to use the measured dry aerosol size distribution and measured dry scattering and absorption coefficients to compute the dry aerosol refractive index using Mie Theory. This dry refractive index is then used with the measured humidified scattering coefficient and Mie Theory to iteratively solve for the effective growth factor, assuming that the humidified aerosol real refractive index is the volume-weighted average of the dry particle refractive index and that of pure water (1.33 - 0i). Sawamura et al. found that the measured in situ aerosol volume and the measured ambient HSRL vertically-resolved retrieval aerosol volume were in excellent agreement once the effective growth factors were used to convert the dry in situ measurements to ambient humidification values. This size invariant growth factor, scaled by scattering coefficient and averaged for each profile, was used to scale the particle sizes for the dried aircraft aerosol volume size distribution to better approximate the same columnar ambient VSD as provided by the AERONET retrievals.

Description of the range of GF values applied during each campaign is now included in Table 2.

Section 4: Ambient relative humidity profiles should be included in the manuscript, stating at least maximum RH encountered in the profiles. This is important to understand the effect of hygroscopic correction, and could be more useful than CWV in Figure 3.

**How the difference of peak radius and width change as a function of maximum or median RH in the profile?**

Average maximum relative humidity for each campaign has been added to Table 1. Plots of vertical profiles of relative humidity for the included AERONET-LARGE comparison plots have been added in Figure 6.

Figure 3 plots have been re-configured to use relative humidity instead of CWV for the xaxis to better emphasize the effects of hygroscopicity.

**Minor comments**

**Page 7, line 3: remove "best quality", I don't think this is necessary. . .**

This has been changed to 'quality-assured'; for readers who may not be familiar with the AERONET Level 2 designation.

**Table 1: +/- standard deviation? State it in the table caption. Also, adjust the number of significant figures according to the +/- value.**

This has been amended.

Page 7, line 10: "... is often correlated with higher relative humidity in these regions due to hygroscopic growth..." -> This sentence, as it is written, it is not a result from this study since this is not clearly seen in Figure 3. In my opinion, it is really speculative, for the MD campaign, there are only 3 data points in Figure 3 (so it is not possible to infer any kind of trend) and for the Texas campaign it is also difficult to see a clear trend of increasing difference with AOD and CWD...

This general statement about the prevalence of hygroscopic growth has been supplemented by a reference regarding hygroscopicity observations during this DISCOVER-AQ campaign.

'From hygroscopicity observations during the Maryland campaign, Ziemba et al. 2013 found that on average, liquid water contributed up to 43% to the ambient extinction coefficient during the study.'

**Page 7, line 13: Totally agree with this statement, but this should be the same for the radius. The association is not clear for neither of them.**

As noted above, we have opted to use profile averaged relative humidity instead of column water vapor as a more direct method of assessing the potential of hygroscopic growth as a source of discrepancy between VSD metrics derived from AERONET and LARGE measurements. The discussion of this analysis has been modified to reflect this in the text.

**Table 2: The average difference in peak radius and width for these same cases but without applying the hygroscopic growth correction should be included in the table for**

**comparison. Table 2 is not directly comparable with Table 1 because of the cases included are different.**

We now include a new table that addresses this. Table 2 provides the AERONET/LARGE comparisons (for only the subset where humidification could be applied) both with and without humidification adjustment applied so the effect of the growth factor correction can be isolated.

**Figure 5: The discussion is focused in the peak radius and width of the size distribution, but looking at Figure 5 there are cases in which the volume concentrations agree well between in-situ and AERONET and others than do not agree. The authors could comment on that.**

The uncertainty of volume concentration for fine mode aerosols is closely related to the uncertainty in the real part of refractive index. AERONET retrievals of the real part of refractive index are sensitive to both measurement noise and instrument offset. This is due to both low retrieval sensitivity to the real part of the refractive index, and due to measurements being affected not only by possible instrumental offsets (such as instrument calibration) but also affected by other somewhat random factors such as atmosphere inhomogeneity.

The variability in the retrieved real part of refractive index is counterbalanced by the variability in the retrieved volume concentration of the fine mode in the AERONET inversion algorithm (Sinyuk et al. 2019, in preparation). The inversion algorithm will reliably provide a highly accurate fit of extinction AOD (within 0.01 at four wavelengths) plus directional sky radiance distributions for each almucantar while introducing a potential additional source of uncertainty to the volume concentration retrievals. Due to somewhat random variability in retrieved real part of refractive index the variability in the retrievals of volume concentration is also random. This may partly explain the good agreement between AERONET retrieved volume concentration and in situ volume concentration in some cases and not in others.

We have expanded the details about retrieval uncertainties in volume concentration in the text.

\*\* Please note that Richard Moore from the NASA LARGE research group has been added as an author to this revised paper

**Intercomparison of aerosol volume size distributions derived from AERONET ground based remote sensing and LARGE in situ aircraft profiles during the 2011-2014 DRAGON and DISCOVER-AQ experiments**

Joel S. Schafer1,3, Tom F. Eck2,3, Brent N. Holben3, Kenneth L. Thornhill4, Luke D. Ziemba4, Patricia Sawamura4, Richard H. Moore4, Ilya Slutsker1,3, Bruce E. Anderson4, Alexander Sinyuk1,3, David M. Giles1,3, Alexander Smirnov1,3, Andreas J. Beyersdorf4,5, and Edward L. Winstead4

1Science Systems and Applications, Inc. Lanham, MD, USA

2Universities Space Research Association, Columbia, MD, USA

3NASA Goddard Space Flight Center, Greenbelt, MD, USA

4NASA Langley Research Center, Hampton, Virginia, USA

[revised manuscript text omitted]

---

## Author Response (AR2)

Reply to Anonymous Reviewer #1:

The papers referenced by Reviewer #1 are primarily dealing with retrieval of size distribution from spectral AOD measurements. In these cases there is just one set of observations to constrain aerosol particles size, volume concentration and the complex index of refraction. To retrieve size distribution, the index of refraction must be fixed *a priori*. Ambiguity in the choice of the index of refraction results in uncertainties which affect both aerosol particles size and volume concentration.

In the case of AERONET however, the spectral measurements of sky radiances provide additional constraints for the aerosol particle size inversion. Thus while variability in the real part of refractive index  results in additional variability in volume concertation this does not affect AERONET retrievals of volume median radius.